# Neurodevelopmental delay and associated factors among preterm infants aged 6 to 24 months adjusted gestation age in two urban hospitals in Uganda

Joyce Nalwoga[1☙], Victoria Nakibuuka[1,2☙], Leonard Manirakiza[3☙],
Tracy Tushabe Namata[1☙], Robert Sebunya[1,4☙]*

**1** Department of Pediatrics, St Francis Nsambya Hospital, Kampala, Uganda, **2** Consultant Neonatologist, St. Francis Hospital Nsambya, Kampala, Uganda, **3** Uganda National Bureau of Standards, Kampala, Uganda, **4** Consultant Pediatric Neurologist, St. Francis Hospital Nsambya, Kampala, Uganda

☙ These authors contributed equally to this work.
* robertsebunya3@gmail.com

## Abstract

Neurodevelopmental delay has been reported among preterm infants who survive beyond the neonatal period. However, there is paucity of data regarding neurodevelopmental outcomes in preterm infants in Africa, including Uganda. This study aimed to determine the prevalence and factors associated with neurodevelopment delay (NDD) in preterm infants aged 6–24 months of adjusted gestation age. A cross-sectional study was conducted among 206 preterm infants, aged between 6 and 24 months of adjusted gestation age in the preterm follow up clinics at St. Francis Hospital Nsambya and Mulago Specialized Women and Neonatal Hospital in Kampala-Uganda from 25th January 2021–25th January 2022. The prevalence of NDD was 13.6% (28/206), with social delay comprising 12.1% (25/206), fine motor delay 11.7% (24/206), language delay 10.7% (22/206) and gross motor delay 7.8% (16/206). Significant factors associated with NDD included microcephaly [aPR = 6.2, CI: 2.6-33.5, P < 0.001], severe acute malnutrition (SAM) [aPR = 4.6, CI: 1.87-12.56, P = 0.021], incomplete immunization [aPR = 2.8, CI: 1.23-4.76, P = 0.013], neonatal sepsis [aPR = 3.8, CI: 1.1-9.3, P = 0.026], neonatal hypoglycemia [aPR = 6.2, CI: 1.8-16.4, P = 0.002], lack of caretaker social support [aPR = 8.3, CI: 2.43-37.9, P = 0.002] and large family size (≥5 children) [aPR = 6.8, CI: 2.24-22.7, P = 0.002]. NDD affects 13.6% of preterm infants, with the social and fine motor delays being most prevalent. Modifiable factors like malnutrition, lack of caretaker social support and incomplete immunization should be screened and addressed to reduce NDD among preterm infants in Uganda.

**Data availability statement:** All relevant data are within the paper and its Supporting Information files.

**Funding:** The author(s) received no specific funding for this work.

**Competing interests:** The authors have declared that no competing interests exist.

## 1. Introduction

According to the World Health Organization, approximately 15 million preterm births occur annually, with 60% of these occurring in Asia and sub-Saharan Africa. Uganda ranks 28th worldwide, with a prematurity rate of 13.6 per 1000 live births [1,2].

Prematurity increases the risk for morbidities such as respiratory distress syndrome, sepsis, necrotizing enterocolitis, intraventricular hemorrhage, seizures, hypoxic ischemic encephalopathy, jaundice, and kernicterus [2,3]. However, as more preterm babies get access to the lifesaving interventions in neonatal units (NNU), their survival rates improve, increasing the burden of NDD, leading to a poor quality of life [4,5].

Developmental delay increases with decreasing gestational age, with an estimated 7% of preterm infants globally surviving with neurodevelopmental impairments [1]. Almost all of the extremely premature neonates have some form of delay and 70% of the severely preterm are delayed at 6 months [2,6]. In a rural district hospital in Eastern Uganda, 20.4% of the preterm infants had NDD in comparison to 7.5% in the term infants [7].

Evidence for increased survival among preterm babies is clearly documented [8], however there is paucity of data regarding developmental outcomes of these preterm infants in different social-economic settings in Uganda. This study therefore aimed at determining the prevalence of neurodevelopmental delay and its associated factors in Ugandan urban hospital settings.

## 2. Methods

### 2.1. Ethics statement

This study was approved by St. Francis Hospital Nsambya Research Ethical Committee (REC number: UG-REC-020)

Written informed consent to participate and publish findings in this study was provided by the participants' legal guardians/next of kin.

### 2.2. Study design

This was a cross-sectional study conducted among preterm infants assessed at 6–24 months adjusted gestation age.

### 2.3. Eligibility criteria

We included all preterm infants aged between 6 and 24 months of adjusted gestation age being followed up at St. Francis Hospital Nsambya and Mulago Specialized Women and Neonatal Hospital, Kampala- Uganda from 25th January 2021–25th January 2022. Preterm infants who were born with congenital anomalies and those with known neurometabolic disorders like myopathies, motor neuron disease were excluded from the study.

### 2.4. Data collection

For the eligible infants, infant and parents' demographics, neonatal and maternal admission history, environmental and parental factors were collected from the parents/guardians using a pre-tested data collection form. The infants' follow up clinic forms were also reviewed for any additional data. This information was fed into a checklist and used to identify the possible factors associated with NDD in the preterm infants.

The infants went through neurodevelopment assessment using the MDAT Screening tool. The infants' head circumference was taken by passing a tape measure around the widest occipito-frontal diameter, weight was measured using Seca balance beam bowls weighing scales for infants that could not stand and for those that could stand using Seca mechanical flat weighing scale, length was measured using a stadiometer. All these were plotted on WHO growth charts as per the children's adjusted age.

Prior to data collection, written informed consent was obtained from the infants' parents.

### 2.5. Statistical analysis

MDAT Z scores were computed online using shiny tool at https://kieran-bromly.shinyapps.io/mdat_scoring_shiny/. The z-scores were computed and assessed by age of the child. For each age category, the z-scores less than -1.64 were categorized as "delayed" while those greater or equal to -1.64 were categorized as "normal". Overall neurodevelopment delay was defined as the total number of infants that were delayed in at least one domain of development.

Infant and neonatal factors, prematurity factors, mother and/or caretaker factors and environmental factors were described using frequency and percentages for the categorical variables and median (interquartile range) for continuous variables. Secondary, Maternal, home/environmental and child descriptive characteristics were compared in bivariate analysis using the t test for 2-sample comparison of continuous data, chi square analysis or fishers exact test for cells less than 5 for multiple-sample comparison of categorical data, and analysis of variance for continuous variable multiple-sample comparisons. Maternal characteristics were compared so that each mother was represented once, avoiding undue weight being given to mothers of multiple gestation. A chi-square test was used to analyze the associations that existed between the development delay measures and associated predictors.

At bivariate analysis, the crude prevalence ratios (cPR) together with their 95% confidence intervals and p-values were presented. All variables with p-values equal 0.2 or below were selected for adjusted multivariate analysis using forward selection technique.

Log binomial analysis was conducted to ascertain the factors associated with neurodevelopmental delay in the gross motor, fine motor, language and social domains; as well as the overall neurodevelopmental delay. Adjusted Prevalence ratios (aPR) together with the 95% confidence intervals and p-values were also presented. All variables with $p \leq 0.05$ were considered to be significant at 5% level of significance.

## 3. Results

### 3.1. Description of neonatal and infant factors

Approximately half of the participants were aged between 6–11 months (n = 98, 47.6%). More than half of the participants were females (n = 109, 52.9%). Majority of the participants' birth weight was 1.5-2.5 kg (n = 127, 61.6%). The head circumference was normal for most of the participants (n = 155, 75.2%) (Table 1).

### 3.2. Gross & fine motor, language, social and over all neurodevelopment delay

The overall neurodevelopmental delay was 13.6% (28/206) (Fig 1). Majority of delays were in the social (25/206) and fine motor (24/206) domains, whilst fewer infants had delay in the language (22/206) and gross motor domains (16/206).

### 3.3. Association between neonatal, infant, environmental & parental factors with overall neurodevelopment delay; bivariate analysis

The significant factors associated with overall neurodevelopment delay included; microcephaly ($X^2$ = 26.592, P < 0.001), male gender ($X^2$ = 5.6105, P = 0.018), SAM ($X^2$ = 8.2117, P = 0.016), neonatal sepsis ($X^2$ = 4.2301, P = 0.04), neonatal hypoglycemia ($X^2$ = 11.70, P = 0.001), ($X^2$ = 8.10, P = 0.004), maternal hypertension ($X^2$ = 3.8715, P = 0.049), having 5 or more

**Table 1. Description of neonatal and infant factors.**

| Variables | Frequency (N = 206) | Percentage (%) |
|---|---|---|
| Grouped Age | | |
| 6 to 11 months | 98 | 47.6 |
| 12 to 17 months | 41 | 19.9 |
| 18 to 24 months | 67 | 32.5 |
| Sex | | |
| Male | 97 | 47.1 |
| Female | 109 | 52.9 |
| Birth weight | | |
| <1 kg | 20 | 9.7 |
| 1 kg to 1.49 kg | 59 | 28.6 |
| 1.5 kg to 2.5 kg | 127 | 61.6 |
| Head circumference | | |
| Microcephaly | 34 | 16.5 |
| Normal | 155 | 75.2 |
| Macrocephaly | 17 | 8.3 |
| Weight for age(current) | | |
| Normal | 121 | 58.7 |
| Under-weight | 85 | 41.3 |
| Weight for Length(current) | | |
| Normal | 178 | 86.4 |
| Moderate Acute malnutrition | 15 | 7.3 |
| Severe acute malnutrition | 13 | 6.3 |
| Length for age(current) | | |
| Normal | 178 | 86.4 |
| Stunted | 28 | 13.6 |
| Immunisation status | | |
| Up to date for age | 176 | 85.4 |
| Incomplete | 19 | 9.2 |
| Complete | 11 | 5.3 |
| Did KMC | | |
| Yes | 163 | 77.7 |
| No | 43 | 22.3 |
| KMC hours | | |
| No KMC | 43 | 20.9 |
| Less than 7 hours | 73 | 35.4 |
| 7 hours and more | 90 | 43.7 |
| Milk used to feed the baby | | |
| Mother's breast milk | 172 | 83.5 |
| Both mother's and donated breast milk | 34 | 16.5 |
| Other feeds | | |
| No | 125 | 60.7 |
| Yes | 81 | 39.3 |
| Child received oxygen when admitted | | |
| Yes | 172 | 83.5 |
| No | 34 | 16.5 |

*(Continued)*

**Table 1.** (Continued)

| Variables | Frequency (N = 206) | Percentage (%) |
|---|---|---|
| Child resuscitated | | |
| Yes | 21 | 10.2 |
| No | 185 | 89.8 |
| Child admitted in last 6 months | | |
| No | 166 | 80.6 |
| Yes | 40 | 19.4 |
| Use of fortifier | | |
| Yes | 101 | 49.0 |
| No | 105 | 51.0 |

siblings ($X^2$ = 12.765, P = 0.005) as well as a caretaker rarely or never getting support from friends and family ($X^2$ = 16.105, P = 0.001).

### 3.4. Factors associated with overall neurodevelopment delay; multivariate analysis

Children with Microcephaly had a prevalence of NDD that was 6.2 times greater than children who had normal head circumference [aPR = 6.2, CI: 2.6-33.5, P < 0.001].

Male children had a prevalence of NDD that was 3.5 times greater than that of their female counterparts [aPR = 3.5, CI: 1.12-7.8, P = 0.034].

Children with SAM had a prevalence of NDD that was 4.6 times greater than that of children who had normal weight-for-length [aPR = 4.6, CI: 1.87-12.56, P = 0.021]. Children who were stunted had a prevalence of NDD that was 3.8 times greater than that of children who had normal length for age [aPR = 3.8, CI: 1.27-6.3, P = 0.043].

Children who were diagnosed with neonatal sepsis had a prevalence of NDD that was 3.3 times greater than that of children who had no neonatal sepsis [aPR = 3.8, CI: 1.1-9.3, P = 0.026]. Well as children who had neonatal hypoglycemia had a prevalence of NDD that was 6.2 times greater than children who did not have hypoglycemia [aPR = 6.2, CI: 1.8-16.4, P = 0.002].

The prevalence of NDD was 2.8 times greater in children with incomplete immunization than in those whose immunization was up to date for age [aPR = 2.8, CI: 1.23-4.76, P = 0.013].

Children that have 5 and more siblings had a prevalence of NDD that was 6.8 times greater than those with less siblings [aPR = 6.8, CI: 2.24-22.7, P = 0.002].

Children whose parents/caretakers rarely got support from their spouses/family/friends had a prevalence of NDD that was 8.3 times higher than those whose parents often got support from their spouses/family/friends [aPR = 8.3, CI: 2.43-37.9, P = 0.002] (Table 2)

### 4. Discussion

#### 4.1. Healthcare Infrastructure and Prevalence of NDD

This study was conducted among preterm infants born and followed up in two urban hospitals which are more equipped with specialist human resource (neonatologists, neonatal fellows, resident/intern doctors and nurses) with competency in the care of preterm neonates. Additionally, these hospitals have modestly well-equipped NICUs and nurseries that allow for fairly adequate management of the complications that come with prematurity as compared to rural hospitals.

In this study, the prevalence of neurodevelopment delay was 13.6%. Various rates have been reported by different authors. Globally, 5% to 52% of infants born preterm in high neonatal mortality rate (NMR) countries survived with long-term neurodevelopment impairments [1]. The prevalence of NDD seen in this study falls within this range.

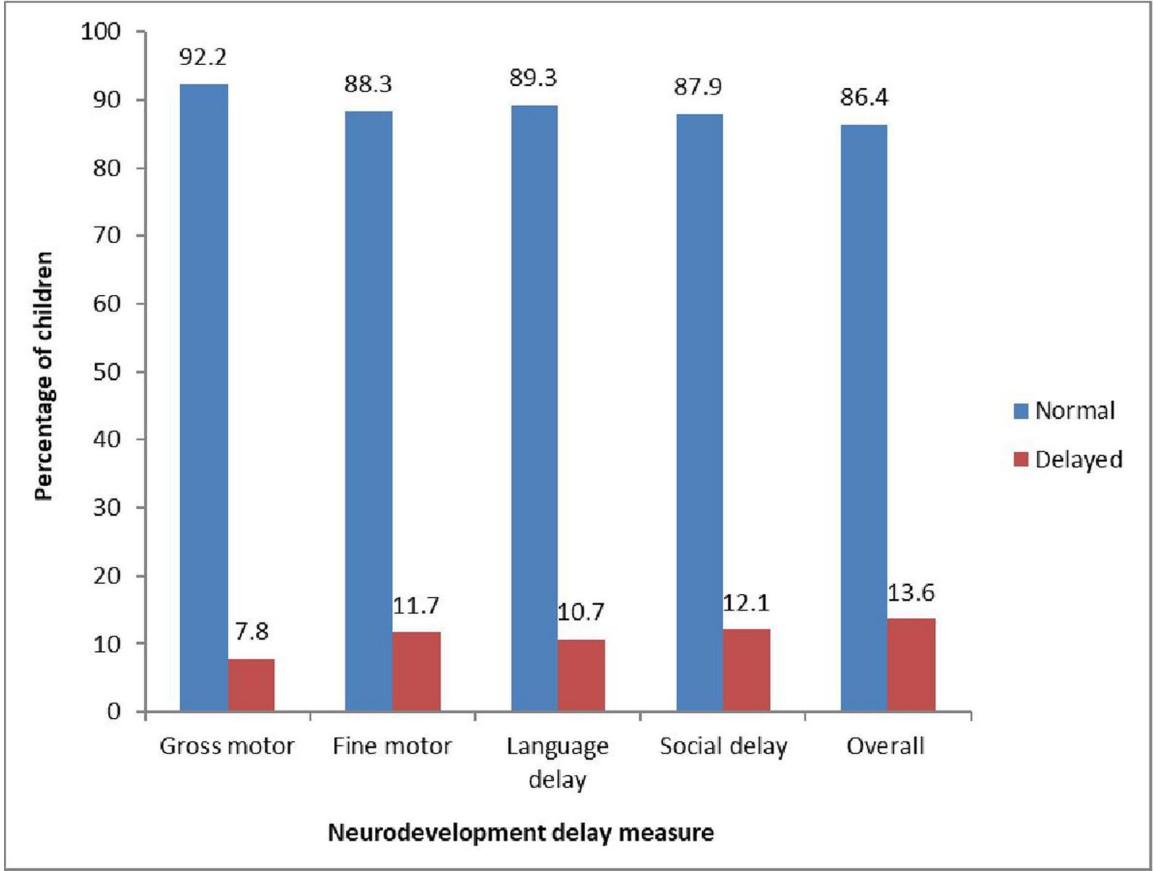

**Fig 1. Gross & fine motor, language, social and over all neurodevelopment delay.**

Conversely, a study done in rural Eastern Uganda (Iganga) found a higher prevalence of NDD in preterm infants born and followed up in a rural hospital (20.4%); especially in the fine motor and language domains [7]. The lower prevalence in our study could be due to the following; firstly, the difference in the healthcare infrastructure/ setting. According to Lutfiyya et al [9], differences in healthcare services offered by rural versus urban settings influence patient disease outcomes. Secondly, much as both studies used the MDAT as the neurodevelopment assessment tool, the method of interpretation was different. This study used Z-scores while the Iganga study used number of failed activities within individual domains to describe NDD.

### 4.2. Key factors associated with NDD among preterm infants

Notably, the common delays amongst these studies were in the social, fine motor and language domains. For proper social, fine motor and language development, ongoing active interaction and stimulation of the infants by their caretakers; as well as intentional exposure to activities around the home that favor development in these domains is needed [10]. The uniform delays in these domains could be attributed (but not limited) to the caretakers' lack of awareness on the need to actively interact with, stimulate and expose the infants to activities that aid in their development.

Microcephaly which was prevalent in 16.5% of this study population is associated with delayed brain growth and reduced brain volume thus impairing neurodevelopment. Preterm infants are at risk of periventricular leukomalacia associated microcephaly [11]. A study in Vietnam demonstrated similar findings wherein infants with low head circumference

**Table 2. Factors associated with overall neurodevelopment delay; multivariate analysis.**

| Variable | Crude analysis | | | Adjusted analysis | | |
|---|---|---|---|---|---|---|
| | cPR | P-value | 95%CI | aPR | P-value | 95% CI |
| **Current Age** | | | | | | |
| 6 to 11 months | 1.0 | – | | – | – | – |
| 12 to 17 months | 0.27 | 0.07 | 0.06 -1.10 | – | – | – |
| 18 to 24 months | 0.65 | 0.28 | 0.30 -1.41 | – | – | – |
| **Gestation age** | | | | | | |
| <28 weeks | 1.0 | – | – | | | |
| 28 to 31 weeks | 1.49 | 0.69 | 0.21-10.7 | – | – | – |
| 32 to 33 weeks | 1.38 | 0.74 | 0.19-9.8 | – | – | – |
| 34 to 37 weeks | 1.31 | 0.78 | 0.19-9.1 | – | – | – |
| **Weight at birth** | | | | | | |
| 1.5kg to 2.5 kg | 1.0 | – | – | | | |
| 1 kg to 1.49 kg | 0.89 | 0.77 | 0.39-2.02 | – | – | – |
| < 1 kg | 1.49 | 0.42 | 0.56-3.99 | – | – | – |
| **Head circumference** | | | | | | |
| Normal | 1.0 | – | – | 1.0 | – | – |
| Microcephaly | 5.32 | <0.0001 | 2.7-10.5 | 6.2 | **<0.001** | 2.6-33.5 |
| Macrocephaly | 1.52 | 0.561 | 0.37-6.22 | 1.8 | 0.113 | 0.11-5.7 |
| **Sex** | | | | | | |
| Female | 1.0 | – | – | 1.0 | – | – |
| Male | 2.37 | 0.023 | 1.13 – 4.99 | 3.5 | **0.034** | 1.12-7.8 |
| **Weight for Length** | | | | | | |
| Normal | 1.0 | – | – | 1.0 | – | – |
| Moderate Acute malnutrition | 1.78 | 0.30 | 0.60-5.3 | 2.1 | 0.165 | 0.11-6.9 |
| Severe acute malnutrition | 3.42 | 0.003 | 1.5-7.6 | 4.6 | **0.021** | 1.87-12.56 |
| **Length for age** | | | | | | |
| Normal | 1.0 | – | – | | 1.0 | – |
| Stunted | 2.1 | 0.05 | 1.0-4.5 | 3.8 | **0.043** | 1.27-6.3 |
| **Neonatal sepsis** | | | | | | |
| No | 1.0 | – | – | 1.0 | – | – |
| Yes | 2.0 | 0.042 | 1.03-4.01 | 3.3 | **0.026** | 1.1-9.3 |
| **Neonatal Jaundice** | | | | | | |
| No | 1.0 | | | – | – | – |
| Yes | 1.02 | 0.95 | 0.50-2.1 | – | – | – |
| **Neonatal Hypoglycemia** | | | | | | |
| No | 1.0 | – | – | 1.0 | | |
| Yes | 4.7 | <0.0001 | 2.3-9.96 | 6.2 | **0.002** | 1.8-16.4 |
| **Immunization status** | | | | | | |
| Up to date for age | 1.0 | – | – | 1.0 | – | – |
| Incomplete | 1.26 | 0.68 | 0.42-3.83 | 2.8 | **0.013** | 1.23-4.76 |
| Complete | 2.2 | 0.14 | 0.77-6.18 | 1.3 | 0.231 | 0.36-9.32 |
| **Received Oxygen therapy** | | | | | | |
| Yes | 1.0 | | | – | – | – |
| No | 1.7 | 0.185 | 0.78-3.65 | – | – | – |

*(Continued)*

**Table 2.** (Continued)

| Variable | Crude analysis | | | Adjusted analysis | | |
|---|---|---|---|---|---|---|
| | cPR | P-value | 95%CI | aPR | P-value | 95% CI |
| **Use of fortifier** | | | | | | |
| No | 1.0 | – | – | | | |
| Yes | 1.20 | 0.61 | 0.60-2.39 | – | – | – |
| **Did KMC** | | | | | | |
| No | 1.0 | – | – | 1.0 | – | – |
| Yes | 3.4 | 0.084 | 0.85-13.9 | 2.1 | 0.104 | 0.76-8.61 |
| **Number of siblings** | | | | | | |
| None | 1.0 | – | – | 1.0 | – | – |
| 1 to 2 siblings | 0.98 | 0.97 | 0.35-2.72 | 0.75 | 0.234 | 0.03-8.32 |
| 3 to 4 siblings | 2.5 | 0.07 | 0.92-6.7 | 3.9 | 0.035 | 1.23-16.34 |
| 5 and above | 5.2 | 0.005 | 1.64-16.5 | 6.8 | **0.002** | 2.24-22.7 |
| **Maternal Hypertension** | | | | | | |
| No | 1.0 | | | 1.0 | | |
| Yes | 2.0 | 0.05 | 1.0 – 3.92 | 1.98 | 0.086 | 0.15 – 7.71 |
| **Perceived support from others** | | | | | | |
| Often | 1.0 | – | – | 1.0 | – | – |
| Sometimes | 1.42 | 0.42 | 0.61-3.3 | 1.64 | 0.376 | 0.023-6.63 |
| Rarely | 6.8 | <0.0001 | 3.3-13.8 | 8.3 | **0.002** | 2.43-37.9 |
| Never | 4.5 | 0.042 | 1.1-19.2 | 5.5 | 0.031 | 1.96-24.81 |

z scores had nearly twice the risk for poor neurodevelopment [12]. Also, in Austria, head circumference catch up growth was associated with better neuromotor scores as compared to failure to attain head circumference catch up growth [13].

Neurodevelopment delay in preterm infants is associated with various challenges for caretakers including but not limited to; increased cost of living, job losses and impeded social activities [14], and consequently increased and sustained levels of stress [15]. Therefore, support systems are needed for these caretakers to enable improved neurodevelopment outcomes for their children because parental stress is associated with poor neurodevelopment among infants [15,16]. Whereas the traditional family model in Uganda often involved large, communal households with extended family members, all offering economic, social and physical support in the care and upbringing of children, there has been an exodus to smaller nuclear families and even single-parent households, especially in urban settings where high costs of living and limited housing spaces among other factors contribute to inability to sustain large extended family households. This consequently leaves the parent (s)/caregivers with limited support and with the sole responsibility of taking care of their children. This study was done in an urban setting where the above may have applied, to the parents/caregivers, and can explain their perceived lack of support from others. Ramona et al 2024 reported 20% of mothers of extreme and very low birth weight (E/VLBW) infants developed depression, anxiety, post-traumatic stress symptoms and parenting stress [17].

Comparable to the study by Nakasone et al [18], male preterm infants were more prone to NDD compared to their female counterparts. The mechanisms for this are not yet fully understood however, placentae of male preterm neonates have been found to have alterations in the pro-oxidant/antioxidant balance with a predominantly pro-oxidant status in their placentae; indicating that males maybe more vulnerable to birth associated oxidative stress [19].

Children with malnutrition (SAM and stunting) were significantly delayed in neurodevelopment. These findings were comparable to findings by Ahishakiye et al [20] in Rwanda and Namazzi et al [7] in Eastern Uganda. Malnutrition a major health hazard in developing countries like Uganda is associated with poor feeding, infections, and poor gut development

and thus exposes preterm babies to high risk of NDD [1]. Failure to initiate complementary feeds and big household number of children may have contributed to the malnutrition in some of the infants in this study.

History of neonatal sepsis and hypoglycemia in the infants was associated with a 3.3 [aPR = 3.8, CI: 1.1-9.3, P = 0.026] and 6.2 [aPR = 6.2, CI: 1.8-16.4, P = 0.002] fold increase in prevalence of NDD respectively. This finding might not be surprising and can be explained by the effects of both sepsis and hypoglycemia on the developing brain. Sepsis-associated brain dysfunction secondary to excessive microglial activation, impaired cerebral perfusion, blood–brain-barrier dysfunction, and altered neurotransmission is a frequent but often neglected occurrence yet it has significant influence on neuro-development [21]. Neonatal hypoglycemia on the other hand is associated with neonatal neuronal damage and death with consequent cognitive and neurodevelopment impairments later in life [22]. Our study findings are similar to a report by Schlapbach et al where proven sepsis independently contributed to neurodevelopment impairment in preterm infants [23].

Children whose immunization status was incomplete were 2.8 times more likely to have neurodevelopment delay compared to their counterparts. This is contrary to Mawson, et al., (2017) who reported that neurodevelopment impairments greatly increased in preterm infants that had received vaccination [24]. Still according to Mawson et al, vaccination associated adverse outcomes like apnea, bradycardia, cardiorespiratory arrest, hepatic encephalopathy can all result in hypoxic ischemic brain injury contributing to neurodevelopment impairment later in life [24]. This finding was especially significant in extreme and very preterm neonates as well as in extremely and very low birth weight pre-terms vaccinated in the very early days of life. The different findings in this study could be because most of the infants in this study were moderate to late and low birth weight pre-terms. Secondly, neonates in our study population often received their first vaccines at discharge or after hospital discharge when they were clinically stable and several days of life old; thereby preventing adverse effects noticed in the above study [24]. Additionally, although the initial vaccination was delayed, infants received subsequent vaccinations in a timely manner and completed their schedules on time. This may have accorded them protection against the immunizable infections that are associated with increased risk for hospitalization and impaired development. The World Health Organization (WHO), due to the high risk of infection in preterm infants, recommends that preterm infants be vaccinated according to chronological age as other infants without correction for gestational age or birth weight; with the exception of hepatitis B vaccination in infants weighing less than 2,000g because of a documented reduced immune response [25].

Infants from families with many children were more prone to NDD compared to their counterparts that belonged to families with fewer children. Kirk et al., (2017) as well noted that having fewer children in the household was associated with better neurodevelopment [26]. Parents/caretakers with fewer children may be in a better position to provide financial, emotional, social and time resources adequately for their preterm infants.

Unlike numerous studies that have shown that the risk of NDD increases with decreasing gestation age and weight at birth [27–29], these factors were not associated with NDD in this study. This may be because the majority of children involved in this study were moderate to late preterms with low (between 1.5kgs to 2.5kgs) and not extremely/very low birth weights.

## 5. Conclusion

NDD affects 13.6% of preterm infants, with the social and fine motor delays being most prevalent. This study demonstrated that NDD in preterm infants is influenced by a spectrum of both modifiable and non-modifiable factors. This spectrum commences in the perinatal period and runs through the neonatal and infant periods involving interactions of medical, nutritional, social and environmental factors. Some of these modifiable factors, e.g., lack of social support systems, malnutrition, neonatal sepsis, neonatal hypoglycemia and incomplete immunization if addressed, could contribute to improved outcomes in these children. Optimizing care for at risk preterm infants may play a role in mitigating the neurodevelopment challenges of being born premature. Future research exploring the impact of early interventions on reduction of NDD will be informative.

## Supporting information

**S1 Data. Raw data sets used to generate results for the study.**
(XLSX)

**S2 Data. Raw data set.**
(CSV)

## Acknowledgments

We would like to thank all the research assistants who rendered their time in collecting data for this study. We would also like to thank all staff members of St. Francis Nsambya Hospital and Mulago Specialized Women and Neonatal Hospital Uganda for allowing us to conduct this study at these facilities. Our sincere appreciation goes to all parents/guardians as well as the infants who participated in this study.

## Author contributions

**Conceptualization:** Joyce Nalwoga, Victoria Nakibuuka, Tracy Tushabe Namata, Robert Sebunya.

**Data curation:** Leonard Manirakiza.

**Formal analysis:** Joyce Nalwoga, Leonard Manirakiza, Tracy Tushabe Namata.

**Investigation:** Joyce Nalwoga, Victoria Nakibuuka, Tracy Tushabe Namata, Robert Sebunya.

**Methodology:** Joyce Nalwoga, Victoria Nakibuuka, Tracy Tushabe Namata, Robert Sebunya.

**Project administration:** Victoria Nakibuuka, Robert Sebunya.

**Software:** Leonard Manirakiza.

**Supervision:** Victoria Nakibuuka, Robert Sebunya.

**Validation:** Victoria Nakibuuka, Robert Sebunya.

**Visualization:** Joyce Nalwoga, Leonard Manirakiza.

**Writing – original draft:** Joyce Nalwoga, Victoria Nakibuuka, Tracy Tushabe Namata, Robert Sebunya.

**Writing – review & editing:** Joyce Nalwoga, Victoria Nakibuuka, Leonard Manirakiza, Tracy Tushabe Namata, Robert Sebunya.

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
