## [Decision Letter · Decision Letter 0]

4 Apr 2025

PGPH-D-25-00368

Neurodevelopmental Delay and Associated Factors among Preterm Infants Aged 6 to 24 months Adjusted Gestation Age in Two Urban Hospitals in Uganda.

Dear Dr. Sebunya,

Thank you for submitting your manuscript to PLOS Global Public Health. After careful consideration, we feel that it has merit but does not fully meet PLOS Global Public Health’s publication criteria as it currently stands. Therefore, we invite you to submit a revised version of the manuscript that addresses the points raised during the review process.

We look forward to receiving your revised manuscript.

Kind regards,

John R Weinstein, PhD, MS

Academic Editor

Journal Requirements:

1. Please provide separate figure files in .tif or .eps format. For more information about figure files please see our guidelines:  LINKhttps://journals.plos.org/globalpublichealth/s/figures https://journals.plos.org/globalpublichealth/s/figures#loc-file-requirements 2. We have noticed that you have uploaded Supporting Information files, but you have not included a list of legends. Please add a full list of legends for your Supporting Information files after the references list. 3. We note that your Data Availability Statement is currently as follows: All data that was generated and used in this research is included in this manuscript and it is available publically.  Please confirm at this time whether or not your submission contains all raw data required to replicate the results of your study. Authors must share the “minimal data set” for their submission. PLOS defines the minimal data set to consist of the data required to replicate all study findings reported in the article, as well as related metadata and methods (https://journals.plos.org/plosone/s/data-availability#loc-minimal-data-set-definition).  For example, authors should submit the following data:  - The values behind the means, standard deviations and other measures reported; - The values used to build graphs; - The points extracted from images for analysis.  Authors do not need to submit their entire data set if only a portion of the data was used in the reported study.  If your submission does not contain these data, please either upload them as Supporting Information files or deposit them to a stable, public repository and provide us with the relevant URLs, DOIs, or accession numbers. For a list of recommended repositories, please see https://journals.plos.org/plosone/s/recommended-repositories.  If there are ethical or legal restrictions on sharing a de-identified data set, please explain them in detail (e.g., data contain potentially sensitive information, data are owned by a third-party organization, etc.) and who has imposed them (e.g., an ethics committee). Please also provide contact information for a data access committee, ethics committee, or other institutional body to which data requests may be sent. If data are owned by a third party, please indicate how others may request data access.

Additional Editor Comments (if provided):

Reviewers' comments:

Reviewer's Responses to Questions

**Comments to the Author**

1. Does this manuscript meet PLOS Global Public Health’s publication criteria ? Is the manuscript technically sound, and do the data support the conclusions? The manuscript must describe methodologically and ethically rigorous research with conclusions that are appropriately drawn based on the data presented.

Reviewer #1: Partly

Reviewer #2: Yes

2. Has the statistical analysis been performed appropriately and rigorously?

Reviewer #1: Yes

Reviewer #2: Yes

3. Have the authors made all data underlying the findings in their manuscript fully available (please refer to the Data Availability Statement at the start of the manuscript PDF file)?

Reviewer #1: Yes

Reviewer #2: Yes

4. Is the manuscript presented in an intelligible fashion and written in standard English?

Reviewer #1: Yes

Reviewer #2: Yes

5. Review Comments to the Author

Reviewer #1: Dear Author,

Kindly find comments/feedback below

Thank you.

ABSTRACT:

For clarity and smooth reading, change

"This study aimed at determining...." to "This study aimed to determine..."

"Factors associated with NDD included microcephaly..." → Consider rewording for clarity, e.g., "Significant factors associated with NDD included microcephaly, severe acute malnutrition (SAM), and incomplete immunization."

"Families with more than 5 children" → "Large family size (≥5 children)" might be more concise.

Statistic: The p-values are presented inconsistently. Some with three decimal places, others with two. Consider formatting them uniformly (e.g., P = 0.002 for all)

Conclusion: Instead of "The 13.6% prevalence of NDD is noteworthy...", consider "NDD affects 13.6% of preterm infants, with social and fine motor delays being most prevalent."

INTRODUCTION;

The citation "1,2" immediately following the statistic on preterm births could be more specific. For example: "According to the World Health Organization, approximately 15 million preterm births occur annually, with 60% of these occurring in Asia and sub-Saharan Africa."

This would clarify the source of the statistic and avoid confusion, especially if the citations are to be included in the references section.

You mentioned that the "7% of preterm infants" and "developmental delay has been shown to increase with decreasing gestational age". You could merge these ideas by saying: "Developmental delay increases with decreasing gestational age, with an estimated 7% of preterm infants globally surviving with neurodevelopmental impairments”

Consider revising this sentence for better flow: "Prematurity increases the risks for morbidities such as respiratory distress syndrome, sepsis, necrotizing enterocolitis, intraventricular hemorrhage, seizures, hypoxic ischemic encephalopathy, jaundice and kernicterus." You could rephrase it as: "Prematurity increases the risk for morbidities such as respiratory distress syndrome, sepsis, necrotizing enterocolitis, intraventricular hemorrhage, seizures, hypoxic ischemic encephalopathy, jaundice, and kernicterus."

DISCUSSION AND CONCLUSION

Breaking down some of the more complex information into bullet points or subheadings (e.g., Key Risk Factors, Healthcare Infrastructure, Social Determinants) would improve readability and help the reader navigate the discussion more easily. While the study compares findings from different regions, it could be helpful to explicitly state at the beginning of the section the main differences between the urban hospital setting in this study and the rural Iganga setting. This could help the reader immediately grasp the key contextual differences before diving into the more technical discussions. The discussion would benefit from specific statistical data or quantitative results from the study to back up claims made in the discussion. For example, when mentioning the increased prevalence of NDD in infants with neonatal sepsis or hypoglycemia, referencing exact percentages or relative risks would make the argument stronger and more grounded in the data. The conclusion summarizes the findings well, but it could be more impactful by focusing on future research directions such as exploring how early interventions could reduce NDD.

There was no report on whether this occurred in first-time mothers or not, child with a sibling that was full-term or not.

Reference

Ensure references are cited properly with correct DOIs and uniform although

Check citation 25.

Minor grammar check to ensure repetitions.

Reviewer #2: 194, 197 : If this question was asked as per 197, last item in the table (sometimes, rarely or never), I would report it as “perceived support” from others given the very subjective nature of responses. Further more, the construct of social support in the context of child care is very variable across settings and I would couch the discussion of findings on this parameter with the deeper explanation of context of the extended family support system and expectations in the setting/ location where this study was conducted so that readers could discern how applicable this would be in their context

216 : The Iganga study looked at a cohort of babies born prematurely at a rural hospital while this study was among babies attending follow up clinic in an urban hospital. Delve into the differences conferred by the two approaches and how this could account for ( or not ) the differences. Also delve into the inherent selection bias and limitations given this study was done in a follow up clinic and largely in a private urban hospital to help readers to give readers a sense of the internal and external validity of the findings respectively.

6. PLOS authors have the option to publish the peer review history of their article (what does this mean? ). If published, this will include your full peer review and any attached files.

**Do you want your identity to be public for this peer review?** For information about this choice, including consent withdrawal, please see our Privacy Policy .

Reviewer #1: No

Reviewer #2: **Yes: ** Bonaventure Ahaisibwe

---

## [Editor Report · Decision Letter 1]

2 May 2025

Neurodevelopmental Delay and Associated Factors among Preterm Infants Aged 6 to 24 months Adjusted Gestation Age in Two Urban Hospitals in Uganda.

PGPH-D-25-00368R1

Dear Dr Sebunya,

We are pleased to inform you that your manuscript 'Neurodevelopmental Delay and Associated Factors among Preterm Infants Aged 6 to 24 months Adjusted Gestation Age in Two Urban Hospitals in Uganda.' has been provisionally accepted for publication in PLOS Global Public Health.

Best regards,

John R Weinstein, PhD, MS

Academic Editor